# Investigating Morphological and Physiological Responses to Stress in *Begonia semperflorens*

**DOI:** 10.3390/ijms26083514

**Published:** 2025-04-09

**Authors:** Julian Ginori, Chi D. Nguyen, Sandra Wilson, Zhanao Deng, Heqiang Huo

**Affiliations:** 1Mid-Florida Research and Education Center, IFAS-University of Florida, Apopka, FL 32703, USA; jginori@gmail.com (J.G.); chi285@ufl.edu (C.D.N.); 2Department of Horticulture, IFAS-University of Florida, Gainesville, FL 32611, USA; 3Crop Transformation Center, Horticulture Science Department, University of Florida, Gainesville, FL 32611, USA; 4Department of Environmental Horticulture, IFAS-University of Florida, Gainesville, FL 32611, USA; sbwilson@ufl.edu; 5Department of Environmental Horticulture, Gulf Coast Research and Education Center, IFAS-University of Florida, Wimauma, FL 33598, USA; zdeng@ufl.edu

**Keywords:** *Begonia semperflorens*, abiotic stress, photoprotection, oxidative stress, stomatal regulation, cuticle thickness, leaf morphology, high light stress, high temperature stress

## Abstract

*Begonia semperflorens*, or wax begonias, are popular ornamental plants often challenged by heat and high light stress in subtropical and tropical regions. This study examined the responses of two stress-tolerant genotypes (FB08-059 and OPGC 5104) and two stress-susceptible genotypes (Cocktail Vodka and Sprint White) under elevated temperature and light conditions. The results revealed significant genotype-dependent differences in stress responses. Stress-tolerant genotypes demonstrated distinct adaptive traits, including thicker cuticles, acute leaf folding, and elevated anthocyanin accumulation, which collectively contributed to enhanced photoprotection and mitigation of cellular damage. Notably, FB08-059 exhibited the most robust stress-adaptive responses, characterized by a 25.83% increase in cuticle thickness and a threefold increase in anthocyanin content under stress conditions. These adaptations effectively limited ROS accumulation and maintained higher Fv/Fm values, thereby sustaining photosynthetic efficiency relative to the other genotypes. In contrast, stress-susceptible genotypes exhibited increased ion leakage, reduced chlorophyll content, and impaired gas exchange rates, reflecting greater oxidative stress and cellular damage. These findings highlight cuticle thickness, anthocyanin accumulation, and leaf folding as key indicators of heat and light stress resilience. This research provides critical insights for breeding programs focused on improving the resilience of wax begonias, supporting the development of heat- and light-tolerant cultivars for sustainable production in stress-prone environments.

## 1. Introduction

The genus *Begonia* comprises over 1800 species, positioning it among the most taxonomically and ecologically diverse groups in the plant kingdom. This diversity is exemplified by a vast spectrum of morphological, ecological, and adaptive traits, predominantly found within tropical and subtropical ecosystems [1,2,3]. The evolutionary plasticity and adaptability of Begonia continue to draw significant attention from botanists, ecologists, and horticulturists. Ongoing discoveries of new species and hybrids underscore its dynamic evolutionary trajectory and emphasize its substantial horticultural and ecological potential [4]. Among the members of this genus, *Begonia semperflorens-colturum*, or wax begonia, holds considerable economic and esthetic value as one of the most widely cultivated bedding plants [5,6,7]. Their vibrant foliage, continuous flowering habits, and resilience to variable urban environments make them valuable in landscaping for roadsides, walkways, and garden beds. In the United States alone, the economic contribution of wax begonias surpassed USD 130 million in 2019, with Florida recognized as a central hub for their production and commercial distribution [8].

Despite their widespread cultivation and popularity, wax begonia exhibits vulnerability to abiotic stresses, particularly elevated temperatures and intense light exposure. These stressors are becoming increasingly prevalent due to global climate change, which poses significant challenges to sustainable horticultural practices. Current climate models project a global temperature increase of 1.5–4.0 °C by the end of the century, alongside a higher frequency of extreme heat events [9]. These environmental changes exacerbate the susceptibility of wax begonias to stress-induced morphological and physiological impairments, including leaf necrosis, reduced vigor, photoinhibition, and a decline in ornamental quality [10,11]. These challenges are particularly pronounced in subtropical and tropical regions, where heat and highlight intensity are pervasive, posing a considerable threat to the sustainable cultivation of wax begonias. Plants have evolved a range of adaptive strategies to mitigate and tolerate abiotic stresses, including both morphological and physiological modifications. At the morphological level, plants respond by thickening the cuticle, which reduces transpirational water loss and provides a protective barrier against excessive radiation. Additionally, leaf folding can serve as an adaptive mechanism to reduce direct light interception and minimize photodamage [12,13]. At the physiological level, both high light intensity and heat stress impose significant constraints on photosynthetic efficiency. Excessive light exposure induces photoinhibition, reducing the efficiency of photosystem II (PSII) and accelerating chlorophyll degradation, which impairs light-harvesting capacity and limits carbon assimilation [14,15]. Simultaneously, elevated temperatures destabilize chlorophyll-protein complexes and promote the breakdown of photosynthetic pigments, further diminishing the plant’s ability to capture and utilize light energy [11]. To counteract these stressors, plants dynamically adjust their chlorophyll and carotenoid composition to optimize light absorption and dissipation, thereby mitigating photodamage and sustaining photosynthetic performance [16,17]. Additionally, the biosynthesis of anthocyanins and other flavonoids serves as a critical photoprotective mechanism, functioning as light filters to prevent excessive radiation from reaching chloroplasts while scavenging reactive oxygen species (ROS) [18,19,20].

The interplay between high light and heat stress also disrupts cellular homeostasis, triggering the overproduction of ROS, including superoxide radicals (O_2_^−^) and hydrogen peroxide (H_2_O_2_) [21,22,23,24]. When ROS accumulation surpasses cellular detoxification capacity, oxidative stress ensues, leading to membrane lipid peroxidation, protein denaturation, and organellar dysfunction, particularly in chloroplasts and mitochondria [21,25]. To counterbalance these deleterious effects, plants upregulate antioxidant defense systems, including enzymatic components such as superoxide dismutase (SOD), catalase (CAT), and peroxidases (POD), which neutralize ROS and maintain cellular redox homeostasis [22,23,26]. Moreover, heat and light stress influence carbon metabolism by altering carbohydrate partitioning and accumulation, with stressed plants often accumulating soluble sugars such as glucose, sucrose, and fructose as osmoprotectants to stabilize cellular structures and sustain metabolic balance [27]. However, prolonged exposure to these stressors disrupts sugar homeostasis, ultimately leading to impaired growth, diminished energy reserves, and compromised reproductive success [10,28,29]. Although these mechanisms have been extensively studied in major agricultural crops such as maize, wheat, and rice, they remain underexplored in ornamental species like wax begonia, creating a critical gap in our understanding of stress adaptation in these plants.

This study seeks to address this knowledge gap by systematically investigating the morphological and physiological responses of four distinct wax begonia genotypes under conditions of elevated temperature and high light intensity. Two experimental genotypes, FB08-059 and OPGC 5104, were selected based on field observations indicating superior stress tolerance. These were compared with two commercially popular cultivars, ‘Cocktail Vodka’ and ‘Sprint White’, thought to be more susceptible to abiotic stress. The research methodology involved a comprehensive assessment of key stress response parameters, including cuticle thickness, leaf folding angles, pigment composition (anthocyanin and chlorophyll content), ROS accumulation, photosynthetic efficiency (Fv/Fm), and gas exchange rates. By integrating these parameters, this study provides a holistic understanding of the interplay between structural, chemical, and physiological traits in mediating stress adaptation. The findings of this research are anticipated to yield significant insights into the genotype-specific mechanisms of stress tolerance in wax begonia. By elucidating these adaptive responses, this study aims to inform breeding programs focused on improving stress resilience, ensuring the sustainable cultivation of wax begonia in increasingly adverse climatic conditions. Moreover, this work contributes to the broader scientific discourse on abiotic stress responses in ornamental plants, offering knowledge that can drive the development of innovative breeding strategies.

## 2. Results

### 2.1. Leaf Thickness, Fresh and Dry Weight of Four Wax Begonia Genotypes Under Shaded and Non-Shaded Conditions

Table 1 depicts the thickness of leaves, which increased significantly when comparing the main effects of shaded to non-shaded leaves across all genotypes (*p* = 2.053 × 10^−7^). However, a two-way ANOVA revealed no significant interaction between genotype and condition (*p* = 0.26), indicating that the increase in total leaf thickness was consistent among the four genotypes in response to the stress condition. Despite this, FB08-059 and OPGC 5104 showed a greater increase in total leaf thickness (12.79% and 14.91%, respectively) compared to ‘Cocktail Vodka’ and ‘Sprint White’ (7.33% and 9.64%, respectively). To further understand the variation in total leaf thickness, mesophyll and cuticle layers were examined. Mesophyll thickness varied significantly by genotype (*p* = 1.955 × 10^−7^), but no significant effect of condition was observed (*p* = 0.3649). In contrast, a significant genotype–condition interaction was detected for cuticle thickness (*p* = 0.00078), suggesting that genotypes responded differently to elevated temperature and light under non-shaded conditions. FB08-059 exhibited the highest increase in cuticle thickness (25.83%), followed by OPGC 5104 (20.04%), while ‘Cocktail Vodka’ and ‘Sprint White’ showed lower increases (15.74% and 16.03%, respectively).

FB08-059 had the most notable cuticle contribution to total leaf thickness, accounting for 53.8% and 61.1% under shaded and non-shaded conditions, respectively. This was significantly higher than ‘Cocktail Vodka’ and ‘Sprint White’, neither of which exceeded a 46.3% cuticle-to-total leaf thickness ratio in either condition (Table 1). For OPGC 5104, the cuticle contributed 50.2% and 47.6% under non-shaded and shaded conditions, respectively. Comparisons of the tolerant genotypes (FB08-059 and OPGC 5104) to the susceptible genotypes (‘Cocktail Vodka’ and ‘Sprint White’) in the non-shaded condition revealed significant differences in both total leaf thickness (*p* = 1.525 × 10^−10^) and cuticle thickness (*p* = 9.081 × 10^−12^). These results suggest that increased cuticle thickness, contributing to total leaf thickness, may be an adaptive response to heat and light stress in the tolerant genotypes.

All plants survived the 41-day experiment, allowing for comprehensive measurements of morphological and physiological parameters. The combined effects of heat and direct sunlight were evident in shoot and root weights (Table 1). All genotypes exhibited a consistent decrease in shoot and root dry weights with no significant genotype–condition interaction (*p* = 0.17 for shoot dry weight; *p* = 0.62 for root dry weight). Under shaded conditions, FB08-059 had the highest shoot fresh weight (357 g) and dry weight (64 g) per plant, followed by OPGC 5104 (284 g and 61 g, respectively). The decrease in water mass under non-shaded conditions was approximately 50% for FB08-059 and OPGC 5104, compared to a 35% decrease for ‘Sprint White’ and ‘Cocktail Vodka’. Despite the higher water loss, FB08-059 maintained substantial fresh (177.09 g) and dry weights (33.86 g) under non-shaded conditions by the end of the study (Table 1). A significant genotype × condition interaction was observed for fresh weight (*p* = 0.001), indicating distinct responses to high light intensity and elevated temperature among genotypes.

### 2.2. Leaf Folding and Stomatal Density of Four Wax Begonia Genotypes Under Shaded and Non-Shaded Conditions

Leaf folding refers to the angle formed by three points: the leaf margin, the petiole-leaf blade junction, and the opposite leaf margin. All four genotypes exhibited leaf folding in response to the direct sunlight and elevated temperature conditions (non-shaded), but the degree of folding varied significantly among them (Figure 1A). A significant genotype–condition interaction effect was observed for leaf folding, indicating that the genotypes responded differently to the shaded vs. non-shaded conditions. Under non-shaded conditions or high-intensity light and elevated temperature, FB08-059 displayed the most pronounced leaf folding, with the leaf angle decreasing to 43.59°, representing a 62.30% change from the original angle (Figure 1A,B). Similarly, ‘Cocktail Vodka’ and OPGC 5104 also showed significant folding under non-shaded conditions, with leaf angle reductions of 49.79% (from 124.31° to 62.41°) and 52.91% (from 117.71° to 55.44°), respectively. In contrast, the ‘Sprint White’ genotype exhibited the least response to direct sunlight, with only a 25.97% reduction in leaf angle (from 142.56° under shaded conditions to 105.31° under non-shaded conditions).

Microscopic observations revealed significant differences in stomatal morphology and arrangement among the genotypes. All four genotypes displayed clustered stomata, a common morphological characteristic of semperflorens-type begonias, with cluster sizes ranging from one to seven stomata per cluster. Statistical analysis demonstrated significant differences in stomatal densities of new leaves between shaded and non-shaded conditions (*p* = 1.91 × 10^−16^) among the four genotypes (Figure 2). Under shaded conditions, FB08-059 and OPGC 5104 had significantly lower stomatal densities (74.10/mm^2^ and 68.47/mm^2^, respectively) compared to their commercial counterparts, ‘Cocktail Vodka’ (91.76/mm^2^) and ‘Sprint White’ (90.02/mm^2^; *p* = 5.77 × 10^−13^). This trend persisted under non-shaded conditions (*p* = 3.11 × 10^−18^), with FB08-059 and OPGC 5104 maintaining significantly lower stomatal densities (85.74/mm^2^ and 79.58/mm^2^, respectively) compared to ‘Cocktail Vodka’ (104.39/mm^2^) and ‘Sprint White’ (101.95/mm^2^). A two-way ANOVA revealed no significant interaction between light availability and genotype in determining stomatal density (*p* = 0.97), indicating that the increase in stomatal density from shaded to non-shaded conditions was consistent across all four genotypes. These results emphasize the uniformity of the stomatal density response to light availability despite genotype-specific differences in absolute stomatal density levels.

### 2.3. Photosynthetic Parameters, Chlorophyll and Anthocyanin Contents of Four Wax Begonia Genotypes Under Shaded and Non-Shaded Conditions

Stomatal conductance, an essential parameter of photosynthetic performance, exhibited significant variation among genotypes and conditions over the study period. Under shaded conditions, ‘Sprint White’ achieved the highest conductance rate (0.36 mol m^−2^ s^−1^), while OPGC 5104 displayed the lowest (0.12 mol m^−2^ s^−1^) (Figure 3A). Despite these differences, a consistent trend emerged across all genotypes in shaded conditions: stomatal conductance declined initially during the first 6–9 days, followed by gradual recovery. Commercial genotypes (‘Sprint White’ and ‘Cocktail Vodka’) consistently exhibited higher stomatal conductance rates under shaded conditions compared to the non-commercial genotypes (FB08-059 and OPGC 5104). In non-shaded conditions, FB08-059 maintained significantly higher stomatal conductance rates than ‘Cocktail Vodka’, OPGC 5104, and ‘Sprint White’ at all time points (Figure 3A). Remarkably, FB08-059 was the only genotype to achieve stomatal conductance levels in the non-shaded condition comparable to those in the shaded condition on day 41 (0.247 mol m^−2^ s^−1^ shaded; 0.234 mol m^−2^ s^−1^ non-shaded) (Figure 3A, Appendix A). A two-way ANOVA revealed a significant interaction between genotype and condition (*p* = 0.0006), underscoring the distinct physiological responses of the genotypes to the experimental conditions.

Transpiration rates mirrored the patterns observed in stomatal conductance. Under shaded conditions, the commercial genotypes (‘Sprint White’ and ‘Cocktail Vodka’) demonstrated higher rates of water loss compared to FB08-059 and OPGC 5104 throughout the study. However, in non-shaded conditions, the transpiration rates of ‘Sprint White’ and ‘Cocktail Vodka’ declined more sharply than those of FB08-059 and OPGC 5104 (Figure 3B). Notably, ‘Sprint White’ consistently exhibited a progressive decrease in transpiration rate under non-shaded conditions. By day 41, FB08-059 showed no significant difference in transpiration rates between shaded (0.00235 mol m^−2^ s^−1^) and non-shaded (0.00207 mol m^−2^ s^−1^) conditions, whereas the commercial genotypes displayed significant reductions in transpiration between conditions (Figure 3B, Appendix A). These results were further supported by a significant genotype–condition interaction effect observed in a two-way ANOVA (*p* = 0.0025).

Carbon assimilation rates, a direct measure of photosynthetic capacity, were consistently higher in shaded plants compared to their non-shaded counterparts, reflecting lower stress levels in shaded conditions (Figure 3C). Under non-shaded conditions, all genotypes initially exhibited suppressed carbon assimilation rates, followed by a gradual recovery over the study period (Figure 3C). On day 41, FB08-059 exhibited a significant difference in carbon assimilation between conditions (12.57 µmol m^−2^ s^−1^ shaded; 10.29 µmol m^−2^ s^−1^ non-shaded), contrasting with its consistent performance in stomatal conductance and transpiration (Figure 3C, Appendix A). The two-way ANOVA revealed a highly significant interaction between genotype and condition (*p* = 1.53 × 10^−5^), indicating substantial variability in carbon assimilation responses among genotypes. Notably, FB08-059 and OPGC 5104 experienced smaller reductions in carbon assimilation under non-shaded conditions compared to the commercial genotypes ‘Cocktail Vodka’ and ‘Sprint White’.

Anthocyanins serve as crucial antioxidants that mitigate oxidative damage induced by abiotic stressors. Leaf pigmentation exhibited a pronounced response to direct sunlight (non-shaded conditions), with all four genotypes developing darker foliage compared to their counterparts grown under shaded conditions (Figure 4A). However, anthocyanin accumulation varied significantly among genotypes. In OPGC 5104 and ‘Sprint White’, both of which possess green foliage, anthocyanin content was negligible under both light conditions (Figure 4A,B). Conversely, in the red-foliaged genotypes, FB08-059 and ‘Cocktail Vodka’, anthocyanin levels were not significantly different under shaded conditions, as determined by Tukey’s HSD test (*p* ≤ 0.05). Upon exposure to direct sunlight, anthocyanin content in FB08-059 increased threefold compared to shaded conditions, whereas ‘Cocktail Vodka’ did not exhibit a significant change in anthocyanin accumulation (Figure 5A,B).

Chlorophyll degradation is a key physiological marker of cellular damage resulting from environmental stress. Under non-shaded conditions, ‘Cocktail Vodka’, ‘Sprint White’, and OPGC 5104 exhibited a significant reduction in total chlorophyll content compared to their shaded counterparts (Figure 4C). Notably, FB08-059 was the only genotype that maintained total chlorophyll content irrespective of light condition (Figure 4C). The observed decline in total chlorophyll content in ‘Cocktail Vodka’, ‘Sprint White’, and OPGC 5104 was primarily attributed to reductions in chlorophyll a, the dominant component of total chlorophyll (Figure 4C,D). Additionally, a significant genotype–by-condition interaction effect was detected for total chlorophyll content (*p* = 3.7 × 10^−9^), chlorophyll a (*p* = 2.5 × 10^−9^), and chlorophyll b (*p* = 0.01), indicating differential photoprotective strategies among the genotypes in response to high light intensity.

### 2.4. Photosynthetic Efficiency and Oxidative Stress Responses of Four Wax Begonia Genotypes Under Shaded and Non-Shaded Conditions

An Fv/Fm ratio between 0.75 and 0.8 is indicative of optimal photosynthetic efficiency, reflecting minimal stress and maximal photochemical activity. Under shaded conditions, no statistically significant variation in Fv/Fm values was observed among the four genotypes (Figure 5A). All genotypes maintained relatively stable Fv/Fm values between 0.7 and 0.75 throughout the duration of the experiment, suggesting mild but non-detrimental stress effects. In contrast, exposure to direct sunlight (non-shaded conditions) resulted in a substantial decline in Fv/Fm values (0.45–0.52) by day 3 across all genotypes (Figure 5B). However, genotypic differences in exposure to high light stress became apparent as time progressed. The putatively stress-tolerant genotypes, FB08-059 and OPGC 5104, exhibited significantly higher Fv/Fm values than the susceptible genotypes, ‘Cocktail Vodka’ and ‘Sprint White’, from day 9 onward in the non-shaded condition (*p* = 0.01). Further assessment of membrane integrity through ion leakage analysis reinforced the observed genotypic disparities in stress tolerance. The non-shaded condition resulted in a markedly higher mean percentage of total ion leakage (23.70%) relative to the shaded condition (15.61%), underscoring the deleterious effects of direct sunlight and elevated temperatures. Ion leakage levels varied significantly among genotypes in both shaded (*p* = 0.0002) and non-shaded (*p* = 5.85 × 10^−6^) conditions. Notably, FB08-059 exhibited the lowest ion leakage, suggesting superior membrane stability compared to the commercial genotypes (Figure 5C). Under non-shaded conditions, ion leakage levels were recorded as follows: FB08-059 (11.91%), OPGC 5104 (20.34%), ‘Cocktail Vodka’ (28.85%), and ‘Sprint White’ (33.72%). Additionally, FB08-059 displayed only a 2.25% increase in ion leakage between shaded and non-shaded conditions, in stark contrast to ‘Sprint White’, which exhibited a 12.23% increase. ‘Cocktail Vodka’ showed an intermediate increase of 9.45% between conditions. These results suggest that FB08-059 demonstrates enhanced resilience to high light intensity and thermal stress compared to the commercially available genotypes.

The regulation of reactive oxygen species (ROS) homeostasis is crucial for sustaining cellular function under environmental stress. Disruptions in ROS equilibrium can lead to oxidative damage and cellular dysfunction. Two primary ROS, superoxide (O_2_·^−^) and hydrogen peroxide (H_2_O_2_), were detected using nitroblue tetrazolium (NBT) and diaminobenzidine (DAB) staining, respectively (Figure 6A,B). A statistically significant increase in ROS accumulation was observed in all genotypes under non-shaded conditions, as determined by Tukey’s HSD test (*p* ≤ 0.05) (Figure 6C,D). FB08-059 exhibited the lowest levels of O_2_·^−^ and H_2_O_2_ accumulation, followed closely by OPGC 5104. Conversely, the commercial genotypes ‘Sprint White’ and ‘Cocktail Vodka’ displayed higher O_2_·^−^ accumulation under both shaded (52.70% and 49.84%, respectively) and non-shaded conditions (80.89% and 71.86%, respectively) (Figure 6C). A similar pattern was evident in H_2_O_2_ accumulation, with ‘Sprint White’ and ‘Cocktail Vodka’ exhibiting higher values under both shaded (44.07% and 39.81%, respectively) and non-shaded conditions (61.85% and 60.21%, respectively) (Figure 6D).

## 3. Discussion

### 3.1. Stomatal Density and Gas Exchange Regulation in Wax Begonia

One of the primary responses observed was an increase in stomatal density under non-shaded conditions across all genotypes, consistent with previous research indicating that high light intensity stimulates stomatal development [30,31]. Stomata play a crucial role in regulating gas exchange by facilitating CO_2_ uptake for photosynthesis while minimizing water loss through transpiration. However, despite the increase in stomatal density, CO_2_ assimilation rates declined under high light and temperature conditions, particularly in the stress-susceptible genotypes. This suggests that stomatal density alone does not determine photosynthetic efficiency; rather, other factors such as cuticle thickness, leaf folding, and oxidative stress regulation play a more substantial role in stress adaptation. A similar increase in stomatal density under high light conditions has been documented in Arabidopsis thaliana, where plants grown under elevated light intensities exhibited greater stomatal development; however, their water-use efficiency varied depending on the genotype’s ability to regulate stomatal aperture [32]. In wheat (*Triticum aestivum*), an increase in stomatal density has been correlated with enhanced photosynthetic performance under moderate stress conditions but was detrimental under prolonged drought due to excessive water loss [33]. Additionally, research in grapevines (*Vitis vinifera*) has demonstrated that cultivars adapted to hot and arid climates maintain lower stomatal density but compensate with increased stomatal size, thereby optimizing water use efficiency [34]. These examples highlight that stomatal density alone does not dictate overall plant performance, as its interaction with stomatal conductance, cuticle properties, and internal water regulation determines stress adaptation.

Plants have evolved trade-offs in stomatal regulation under environmental stress. Increased stomatal density can enhance photosynthetic rates in favorable conditions but may lead to excessive water loss under drought and heat stress [35]. In our study, ‘Cocktail Vodka’ and ‘Sprint White’ exhibited significantly lower stomatal conductance under non-shaded conditions than FB08-059, indicating greater stomatal closure as a water conservation strategy. This may explain their reduced transpiration and carbon assimilation rates compared to the tolerant genotypes, which maintained higher stomatal conductance. This pattern is also evident in corn (*Zea mays*), where drought-tolerant hybrids maintain stomatal conductance despite lower stomatal density, optimizing carbon assimilation while preventing excessive water loss [36]. Similarly, in rice (*Oryza sativa*), cultivars with greater stomatal density exhibited higher transpiration rates, but their photosynthetic efficiency was largely dependent on water availability [37].

Transpiration rates also varied significantly among genotypes. While ‘Cocktail Vodka’ and ‘Sprint White’ exhibited a sharp decline in transpiration under non-shaded conditions, FB08-059 maintained relatively stable transpiration rates, indicating an improved capacity for evaporative cooling. This balance between transpiration and stomatal conductance may contribute to the superior stress tolerance of FB08-059 [38]. This relationship has been observed in sunflowers (*Helianthus annuus*), where drought-resistant varieties show lower transpiration rates but maintain sufficient stomatal opening for sustained photosynthesis, allowing them to endure extended periods of heat stress [39]. In contrast, more sensitive varieties experience rapid stomatal closure under high temperatures, leading to reductions in gas exchange and growth rates. Similarly, in cotton (*Gossypium hirsutum*), heat-tolerant genotypes exhibit higher transpiration rates, enabling them to dissipate excess heat and maintain lower leaf temperatures [40].

### 3.2. Structural Adaptations: Leaf Folding and Cuticle Thickening in Wax Begonia

Leaf folding is an adaptive mechanism that reduces light interception and minimizes photodamage under excessive radiation [41]. In this study, FB08-059 displayed the most pronounced leaf folding under non-shaded conditions, followed by OPGC 5104. Leaf folding limits direct light penetration, thereby reducing the risk of photoinhibition and excessive heat buildup [42]. In contrast, ‘Sprint White’ exhibited minimal leaf folding, which may explain its greater susceptibility to photoinhibition and oxidative stress. Leaf folding has been reported as a photoprotective response in various plant species. In cotton (*Gossypium hirsutum*), leaves subjected to high light intensity exhibit enhanced folding to limit radiation absorption and maintain optimal photosynthetic rates [40]. In tropical grasses, leaf rolling, a more extreme form of leaf folding, is a well-documented response to heat and drought stress, helping to conserve moisture and reduce excessive transpiration [43]. These findings suggest that structural modifications, such as leaf folding, contribute significantly to stress mitigation across diverse plant taxa.

Cuticle thickening was another significant morphological response observed in this study. FB08-059 exhibited the highest increase in cuticle thickness under non-shaded conditions, followed by OPGC 5104. A thicker cuticle serves multiple protective functions, including reducing transpirational water loss, enhancing leaf reflectance, and providing a physical barrier against excessive light penetration [44]. Previous studies have also shown that cuticle thickness is correlated with drought and heat tolerance in multiple plant species, including corn and Arabidopsis [12,13]. The greater cuticle thickness in FB08-059 and OPGC 5104 likely contributed to their ability to maintain higher photosynthetic efficiency under stress conditions. Similar responses have been documented in numerous plant species. In wheat, genotypes with thicker cuticles exhibit greater resistance to heat-induced oxidative stress and maintain better water retention than those with thinner cuticles [45]. In rice, increased cuticle deposition has been linked to improved drought tolerance, reducing non-stomatal water loss while simultaneously protecting against leaf desiccation [46]. These findings further reinforce the importance of cuticle thickening as a conserved trait among plant species adapting to extreme environmental conditions.

The combination of leaf folding and cuticle thickening provides an integrated strategy for mitigating abiotic stress. While leaf folding reduces direct light exposure, cuticle thickening enhances overall structural integrity, allowing plants to tolerate prolonged periods of heat and light stress. The differential responses observed among the genotypes in this study indicate that FB08-059 and OPGC 5104 possess superior adaptive traits, which could be valuable for breeding programs aimed at improving stress resilience in ornamental and crop species.

### 3.3. Anthocyanin Accumulation and Photoprotection in Wax Begonia

Anthocyanins have been widely recognized for their role in photoprotection and oxidative stress mitigation. The results of this study demonstrated a significant increase in anthocyanin accumulation in FB08-059 under non-shaded conditions, whereas ‘Cocktail Vodka’ did not exhibit a comparable increase. This suggests that the genetic capacity to upregulate anthocyanin production in response to environmental stress is a key determinant of heat and light tolerance [47,48]. The protective role of anthocyanins is twofold: (1) they act as light filters that absorb excess radiation, thereby reducing potential damage to the photosynthetic apparatus, and (2) they function as antioxidants that neutralize reactive oxygen species (ROS) [18]. The elevated anthocyanin levels in FB08-059 corresponded with lower ROS accumulation and reduced ion leakage, further supporting the hypothesis that anthocyanins play a crucial role in stress tolerance. Anthocyanin accumulation has been documented as a protective mechanism in multiple plant species exposed to high light intensity, reinforcing the role of this pigment in photoprotection [49].

Anthocyanin biosynthesis has been shown to be a critical photoprotective strategy in a variety of plant species. In corn, high light intensity induces anthocyanin accumulation in leaf tissue, leading to improved tolerance against photodamage and heat stress [50]. Similarly, in apples (*Malus domestica*), anthocyanin accumulation in response to high light conditions enhances both photosynthetic efficiency and ROS scavenging, reducing oxidative damage to cellular structures [51]. In Arabidopsis, mutants deficient in anthocyanin biosynthesis exhibit increased susceptibility to high light stress, further emphasizing the importance of this pigment in photoprotection [52]. Additionally, anthocyanin accumulation has been linked to improved drought tolerance in crops such as wheat and rice. Under limited water availability, anthocyanins not only protect against excessive light damage but also enhance osmotic regulation, allowing plants to maintain cellular hydration [53,54]. The findings in FB08-059 align with these studies, suggesting that anthocyanins confer a multifaceted advantage by protecting against both high light and oxidative stress while maintaining physiological stability. Interestingly, in ‘Cocktail Vodka’, the absence of a significant anthocyanin increase suggests that commercial breeding selections may prioritize esthetic consistency over environmental responsiveness. This highlights the potential trade-off between ornamental value and stress resilience, suggesting that future breeding efforts should consider incorporating genetic diversity in anthocyanin regulation to enhance stress tolerance.

### 3.4. Oxidative Stress and Membrane Stability in Wax Begonia

Exposure to high light and temperature conditions induces oxidative stress, leading to ROS accumulation and membrane damage. In this study, FB08-059 and OPGC 5104 exhibited significantly lower ROS accumulation and ion leakage compared to ‘Cocktail Vodka’ and ‘Sprint White’. This indicates that these genotypes possess more efficient ROS scavenging mechanisms, which help maintain membrane stability and overall plant health under stress conditions [24]. The detrimental effects of ROS accumulation have been widely studied in various plant species. In wheat, genotypes with higher levels of enzymatic antioxidants, such as superoxide dismutase (SOD) and catalase (CAT), exhibit greater resilience to heat stress by mitigating oxidative damage [55]. Similarly, in rice, plants with upregulated ascorbate peroxidase (APX) activity maintain superior membrane stability and reduced ion leakage when subjected to high-temperature stress [26]. The ability to efficiently detoxify ROS is, therefore, a crucial determinant of stress tolerance across diverse plant taxa.

Chlorophyll fluorescence (Fv/Fm) measurements further reinforced these findings. While all genotypes experienced a decline in Fv/Fm under non-shaded conditions, FB08-059 and OPGC 5104 maintained significantly higher values than the stress-susceptible genotypes, suggesting a lower degree of photoinhibition. Similar trends have been reported in sunflower (*Helianthus annuus*), where drought-tolerant cultivars exhibit higher Fv/Fm values under water stress due to enhanced protective mechanisms in the photosynthetic apparatus [14]. In corn, genotypes with improved ROS scavenging capacity show sustained Fv/Fm ratios under combined drought and heat stress, allowing for greater retention of photosynthetic efficiency [56]. Beyond enzymatic ROS detoxification, non-enzymatic antioxidants such as flavonoids, carotenoids, and tocopherols also play an essential role in oxidative stress mitigation. In Arabidopsis, flavonoid-deficient mutants display increased susceptibility to high light stress, underscoring the importance of these compounds in photoprotection [54]. Similarly, in grapevines, cultivars with higher carotenoid content exhibit reduced oxidative damage and improved physiological stability under prolonged heat stress [57]. The significant differences in ROS accumulation and ion leakage observed in this study suggest that FB08-059 and OPGC 5104 may possess enhanced antioxidant systems, contributing to their superior stress tolerance.

The interplay between oxidative stress regulation and membrane stability is a key determinant of plant resilience. The observed differences in ROS accumulation, ion leakage, and Fv/Fm values among the genotypes indicate that FB08-059 and OPGC 5104 employ more robust protective mechanisms compared to ‘Cocktail Vodka’ and ‘Sprint White’. These findings align with previous research in multiple crop and ornamental species, highlighting the importance of integrating ROS detoxification pathways into breeding programs aimed at improving stress tolerance.

## 4. Materials and Methods

### 4.1. Plant Material and Growing Conditions

Four wax begonia (*Begonia semperflorens-coltorum*) genotypes were selected for this study: two bronze-foliaged genotypes (FB08-059 and ‘Cocktail Vodka’) and two green-foliaged genotypes (OPGC 5104 and ‘Sprint White’). The plants were propagated from cuttings and cultivated for ten weeks in a low-light greenhouse environment (photosynthetically active radiation, PAR = 750 µmol m^−2^ s^−1^) under a controlled temperature regime of 28/23 °C (day/night). FB08-059, previously identified as heat-tolerant [58], exhibits dark green to red foliage in shaded environments. OPGC 5104 is a wild-type begonia with bright green foliage, originally collected from Hawaii. ‘Sprint White’ and ‘Cocktail Vodka’ are commercially available genotypes with green and red foliage, respectively (Figure 7). Two weeks prior to initiating the experimental conditions, uniform plugs were transplanted into 1-gallon pots containing a bark- and peat-based soilless medium (Premium Nursery and Veg Mix, Reliable Peat Company, Leesburg, FL, USA), supplemented with 5 g/L of 14N-14P-14K slow-release fertilizer (Osmocote, The Scotts Company, Marysville, OH, USA). Plants were then randomly assigned to either a shaded or non-shaded condition under natural sunlight at the Mid-Florida Research and Education Center in Apopka, FL, USA. The average day/night temperatures and humidity are presented in Appendix A. The non-shaded condition was characterized by peak PAR values of 2100 µmol m^−2^ s^−1^, with day/night temperatures of 35/22.5 °C. Shading was achieved using a black shade net (Agfabric, Home Depot, Atlanta, GA, USA), which maintained a peak PAR of 750 µmol m^−2^ s^−1^ and temperatures of 30/22.5 °C. Relative humidity levels fluctuated between 65% and 100% in both conditions. To ensure that drought stress did not confound the results, plants were regularly irrigated via rainfall or an overhead sprinkler system.

### 4.2. Morphological Measurements

Morphological assessments were conducted at the conclusion of the 41-day experiment. Measurements were taken from the first, second, and third fully expanded leaves from the apical meristem. Stomatal density and cluster formation were determined using the clear nail varnish impression method [59]. Stomatal impressions were obtained from the abaxial leaf surface and examined under a light microscope. Stomatal density was calculated by averaging counts from three leaves per plant and normalizing per unit area. To evaluate anatomical differences, leaf cross-sections were obtained using a microtome blade. Sections were examined under a calibrated light microscope to determine total leaf thickness and cuticle thickness. Measurements were restricted to the lamina, excluding primary and secondary veins.

Leaf folding was quantified using ImageJ Version 1.51 software (NIH, Bethesda, MD, USA). Leaves were laid flat with the petiole oriented upwards. Images were captured from a standardized distance, and the angle between the leaf margins and petiole was measured. Each measurement was replicated three times per sample. Shoot and root biomass were measured following the 41-day stress conditions. Above-ground fresh weights were recorded immediately after excising plants at the base of the stem. Dry weights for shoot and root tissues were determined after oven drying at 50 °C for 72 h.

### 4.3. Physiological Measurements

Photosynthetic parameters, including carbon assimilation, stomatal conductance, and transpiration, were recorded throughout the 41-day experiment using a LI-COR 6800 Portable Photosynthesis System (LI-COR Biosciences, Lincoln, NE, USA). Measurements were taken during peak daylight hours (12:00–16:00 h) under the following chamber conditions: relative humidity 40–50%, photosynthetic photon flux density (PPFD) 1000 µmol m^−2^ s^−1^, reference CO_2_ concentration 400 ppm, flow rate 500 µmol s^−1^, and fan speed 10,000 rpm. Following stabilization, three consecutive measurements were recorded for each plant at 30 s intervals.

Anthocyanin and chlorophyll contents were quantified using the protocols described by Taghavi and Porra et al., respectively [60,61]. Leaf samples were collected from each plant at the conclusion of the experiment. Three lamina disks were excised per sample, ensuring the exclusion of major veins. Samples were weighed and processed for absorbance readings using a Synergy H1 microplate spectrophotometer (BioTek, Winooski, VT, USA). Pigment concentrations were calculated using the following equations:$$Anthocyanin\;content=\frac{(A_{530}-A_{657})\times 1000}{mg\;tissue}$$$$Chlorophyll\;a\;and\;b\;Content=\frac{(\left(22.12\times A_{652}\right)+\left(2.71\times A_{665}\right))\times 0.5}{mg\;tissue}$$

### 4.4. Measurements of Stress Responses

Chlorophyll fluorescence (Fv/Fm) was measured using an OS30p chlorophyll fluorometer (Opti-Sciences, Hudson, NH, USA). To ensure dark adaptation, measurements were taken two hours before dawn on three fully expanded leaves per plant. Readings were averaged to determine mean Fv/Fm values.

Electrolyte leakage was quantified as an indicator of membrane stability. Three leaf disks from each sample were submerged in 5 mL deionized water for four hours, and initial conductivity readings were taken using an Orion Star A215 conductivity meter (Thermo Fisher Scientific, Waltham, MA, USA). Samples were then autoclaved at 121 °C for 20 min, allowed to cool for 15 min, and final conductivity readings were obtained. Electrolyte leakage was calculated as: (initial conductivity/final conductivity) × 100. Reactive oxygen species (ROS) accumulation was assessed using diaminobenzidine (DAB) and nitroblue tetrazolium (NBT) staining [62]. Leaves were collected at the end of the study and bisected along the midrib. One half was stained with NBT to detect superoxide radicals (O_2_·^−^), while the other half was stained with DAB to visualize hydrogen peroxide (H_2_O_2_) accumulation. High-anthocyanin samples (FB08-059 and ‘Cocktail Vodka’) required ethanol washes for accurate staining observation.

### 4.5. Statistical Analysis

The experimental design consisted of a completely randomized block arrangement with three replications per block and four plants per genotype in each condition. Trait means and standard errors (n = 12) were calculated at each time interval. At the conclusion of the experiment (day 41), a two-way analysis of variance (ANOVA) was performed to evaluate the effects of genotype and light condition, and their interaction. Significant differences were determined using Tukey’s honestly significant difference (HSD) test (*p*-value ≤ 0.05) in R Studio Version 3.6.0 (Boston, MA, USA) with the Agricolae package. Paired Student’s *t*-tests were conducted to assess significant differences between putatively tolerant and susceptible genotypes. Leaf folding images were captured with a Nikon digital camera and analyzed using ImageJ public domain software. ROS-stained leaf images were processed with GIMP (version 2.10) before quantification in ImageJ.

## 5. Conclusions

The findings of this study showcase key morphological and physiological traits associated with heat and high light stress tolerance in wax begonia. Specifically, increased cuticle thickness, enhanced anthocyanin accumulation, and acute leaf folding were identified as critical adaptive traits. These insights can be directly applied to breeding programs aimed at developing more resilient ornamental cultivars for commercial and landscaping applications. FB08-059, in particular, demonstrated the most robust stress-adaptive responses, making it a valuable candidate for breeding initiatives focused on enhancing heat and light tolerance in wax begonias. Future research should explore the genetic regulation of these adaptive traits and evaluate their heritability to facilitate targeted breeding strategies.

## Figures and Tables

**Figure 1 ijms-26-03514-f001:**
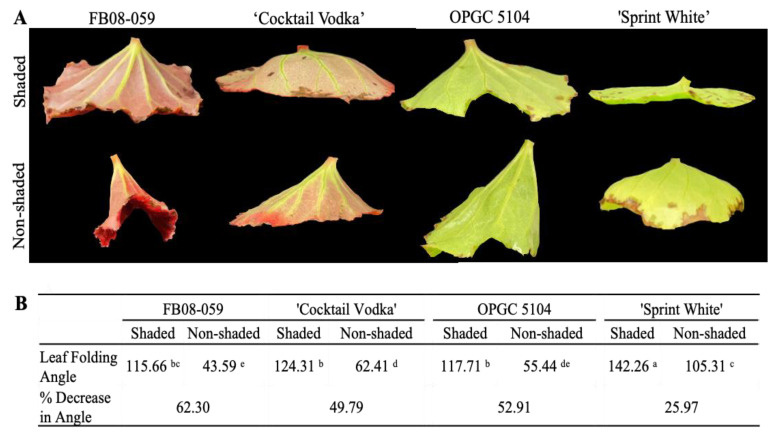
Variations in Leaf Folding Among Four Wax Begonia Genotypes Under Shaded and Non-Shaded Conditions. (**A**) Images depicting the leaf folding patterns of four genotypes in shaded (top) and non-shaded (bottom) conditions. (**B**) Table summarizing the degree (°) of leaf folding for each genotype under shaded and non-shaded conditions. Means within row marked by different letters indicate significant differences according to Tukey’s HSD test (*p* ≤ 0.05). Leaf angles were averaged from three leaves per plant (*n* = 12).

**Figure 2 ijms-26-03514-f002:**
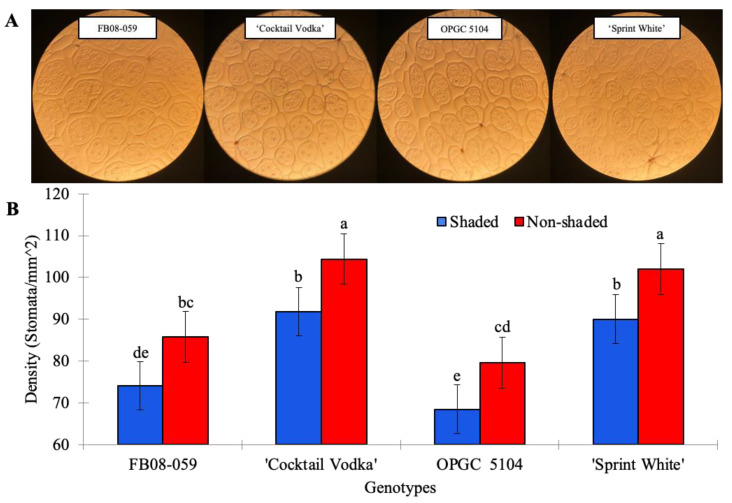
Stomatal Morphology and Density in Four Wax Begonia Genotypes Under Shaded and Non-Shaded Conditions. (**A**) Representative images showing clustered stomata in the four genotypes under shaded conditions. (**B**) Stomatal density for each genotype under shaded and non-shaded conditions. Bars represent the mean ± standard error (n = 12). Letters above bars indicate significant differences across all conditions and genotypes determined by Tukey’s HSD test (*p*-value ≤ 0.05).

**Figure 3 ijms-26-03514-f003:**
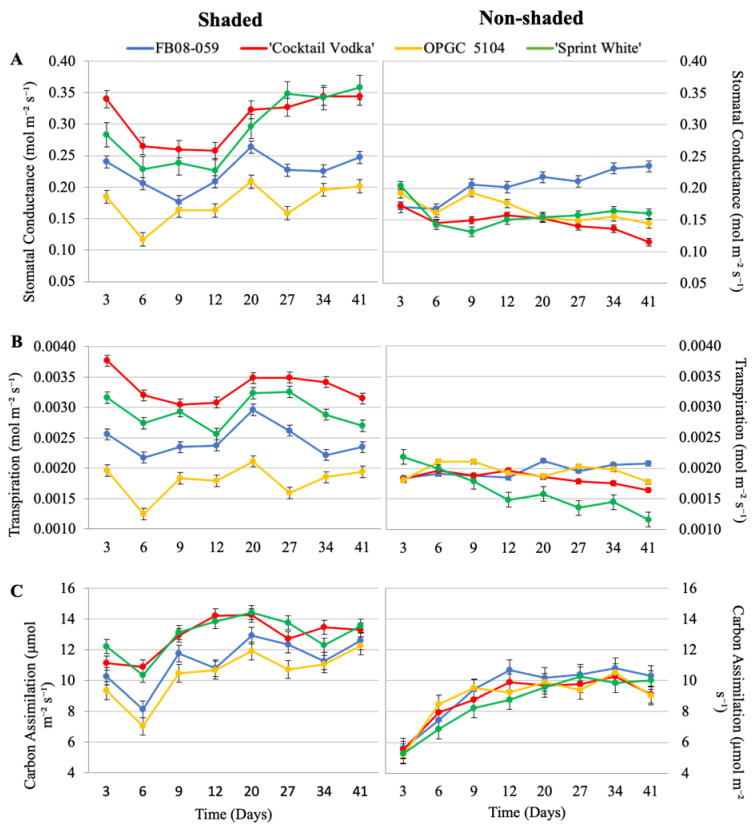
Stomatal Conductance, Transpiration, and Carbon Assimilation Rates of Four Wax Begonia Genotypes Under Shaded and Non-Shaded Conditions. Stomatal conductance (**A**), transpiration (**B**), and carbon assimilation (**C**) responses of wax begonia genotypes (FB08-059, OPGC 5104, ‘Sprint White’, and ‘Cocktail Vodka’) grown under shaded (**left**) and non-shaded (**right**) conditions for 41 days. Measurements were taken during peak light and heat intensity between 12 PM and 3 PM. Bars represent mean ± standard error (n = 12).

**Figure 4 ijms-26-03514-f004:**
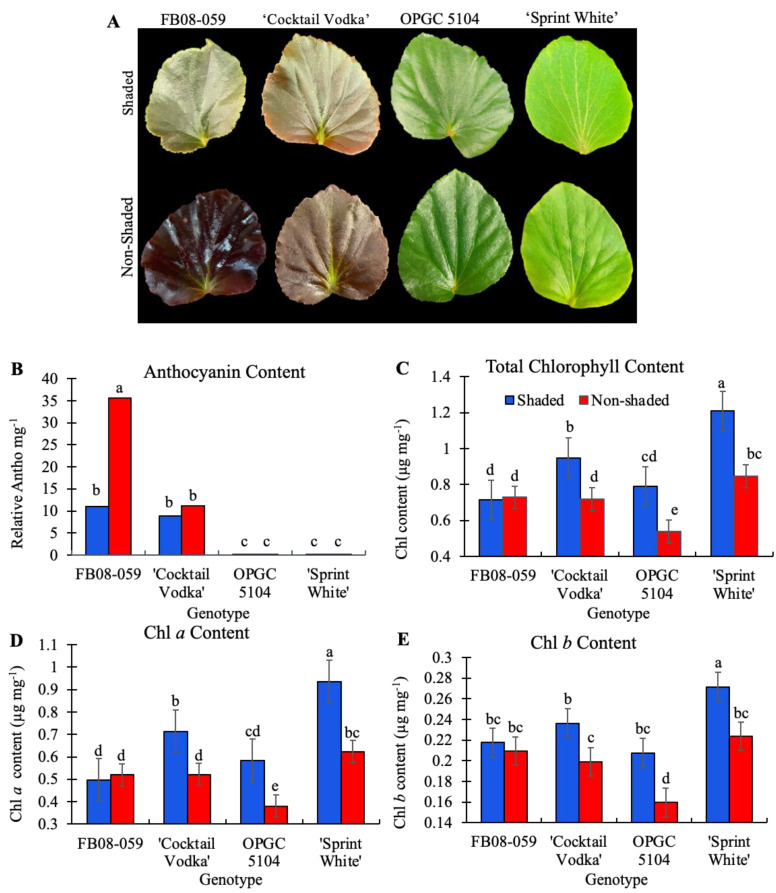
Anthocyanin and Chlorophyll Content in Wax Begonia Genotypes Under Shaded and Non-Shaded Conditions. (**A**) Representative leaf images from each genotype, taken 41 days after exposure to shaded and non-shaded conditions. (**B**) Relative anthocyanin content of four genotypes under shaded and non-shaded conditions. ‘OPGC5104’ and ‘Sprint White’ bars appear minimal due to their negligible anthocyanin levels (<0.1 absorbance units) but are shown to highlight differences. (**C**) Total chlorophyll content (Chl a + Chl b) in four genotypes under shaded and non-shaded conditions. (**D**) Chlorophyll a content. (**E**) Chlorophyll b content. Bars represent mean ± standard error (n = 12). Different letters above bars indicate significant differences according to Tukey’s HSD test (*p*-value ≤ 0.05).

**Figure 5 ijms-26-03514-f005:**
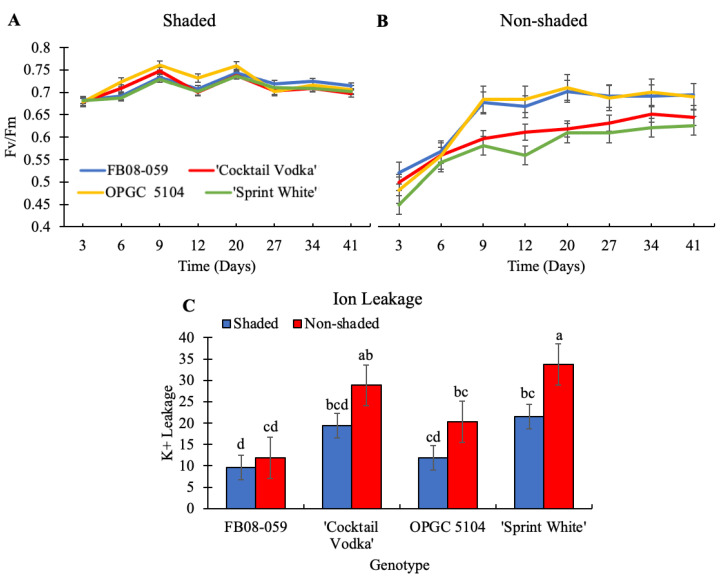
Effects of Shaded and Non-Shaded Conditions on Fluorescence and Electrolyte Leakage in Wax Begonia Genotypes. (**A**) Temporal dynamics of Fv/Fm under shaded conditions across four genotypes. (**B**) Temporal dynamics of Fv/Fm under non-shaded conditions. (**C**) Electrolyte leakage, expressed as the percentage of total ions released. Bars represent mean ± standard error (n = 12). Letters above bars indicate significant differences among all genotypes and conditions based on Tukey’s HSD test (*p*-value ≤ 0.05).

**Figure 6 ijms-26-03514-f006:**
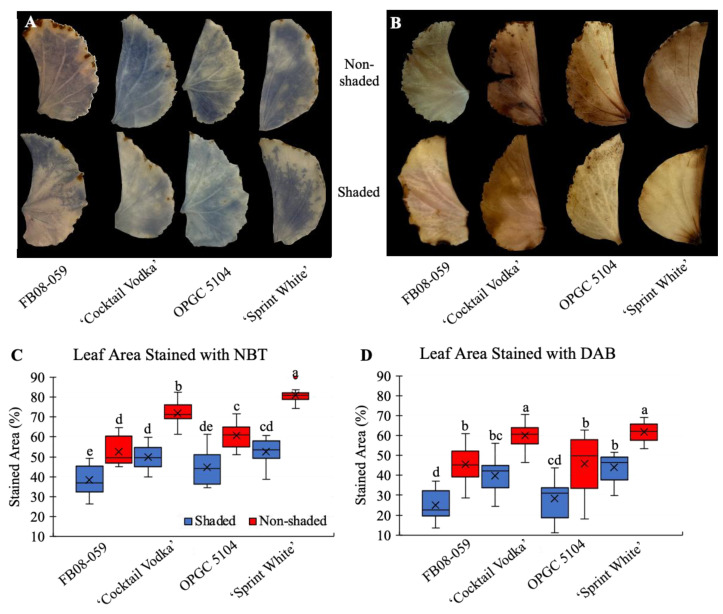
Oxidative Stress Responses in Wax Begonia Genotypes Under Shaded and Non-Shaded Conditions (**A**) NBT staining for superoxide (O_2_·^−^) accumulation in leaf tissue of four genotypes under shaded and non-shaded conditions. (**B**) DAB staining for hydrogen peroxide (H_2_O_2_) accumulation in leaf tissue under both conditions. (**C**) Quantification of O_2_·^−^ accumulation. (**D**) Quantification of H_2_O_2_ accumulation. Bars represent mean ± standard error (n = 12). Different letters indicate significant differences among genotypes as determined by Tukey’s HSD test (*p*-value ≤ 0.05).

**Figure 7 ijms-26-03514-f007:**
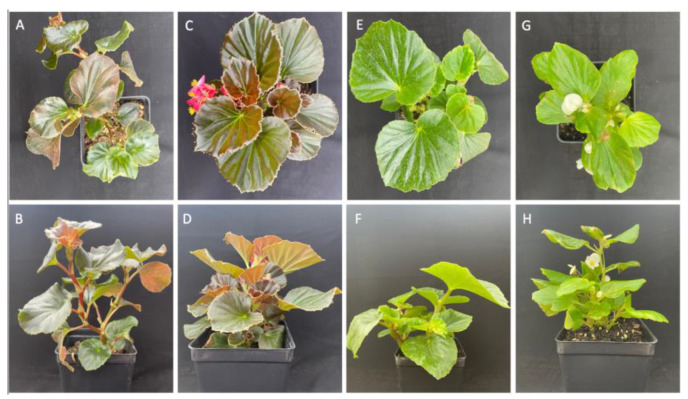
Representative top and side views of four wax begonia genotypes used in this study. FB08-059 (**A**,**B**), ‘Cocktail Vodka’ (**C**,**D**), OPGC 5104 (**E**,**F**), and ‘Sprint White’ (**G**,**H**). Images were captured ten weeks post-cutting propagation, prior to exposure to experimental light conditions. The study examined stress tolerance under shaded and non-shaded conditions.

**Table 1 ijms-26-03514-t001:** Morphological and Physiological Parameters of Four Begonia semperflorens Genotypes Under Shaded and Non-Shaded Conditions.

	FB08-059		“Cocktail Vodka”		OPGC 5104		“Sprint White”	
	Shaded	Non-Shaded	Shaded	Non-Shaded	Shaded	Non-Shaded	Shaded	Non-Shaded
Total Leaf Thickness	5.35 ± 0.09 ^b^	6.04 ± 0.08 ^a^	4.36 ± 0.20 ^d^	4.68 ± 0.12 ^cd^	4.59 ± 0.09 ^d^	5.22 ± 0.10 ^bc^	3.15 ± 0.15 ^e^	3.46 ± 0.08 ^e^
Cuticle Thickness	2.89 ± 0.06 ^b^	3.63 ± 0.10 ^a^	1.86 ± 0.09 ^d^	2.16 ± 0.05 ^cd^	2.18 ± 0.06 ^c^	2.61 ± 0.06 ^b^	1.3 ± 0.06 ^e^	1.51 ± 0.05 ^e^
Mesophyll Thickness	2.47 ± 0.05	2.41 ± 0.11	2.50 ± 0.13	2.53 ± 0.09	2.41 ± 0.07	2.6 ± 0.11	1.85 ± 0.11	1.95 ± 0.08
Cuticle/Total Leaf Ratio	0.54 ± 0.01 ^b^	0.61 ± 0.02 ^a^	0.43 ± 0.01 ^de^	0.46 ± 0.01 ^cde^	0.48 ± 0.01 ^cd^	0.50 ± 0.01 ^bc^	0.42 ± 0.01 ^e^	0.43 ± 0.02 ^de^
Shoot FW (g)	356.87 ± 20.85 ^a^	177.09 ± 16.60 ^cd^	180.53 ± 9.55 ^cd^	109.91 ± 9.74 ^e^	284.11 ± 11.78 ^b^	135.61 ± 7.09 ^cde^	194.68 ± 15.82 ^c^	119.08 ± 13.95 ^de^
Shoot DW (g)	64.31 ± 1.13 ^a^	33.86 ± 1.96 ^c^	52.14 ± 0.49 ^b^	25.07 ± 1.39 ^d^	61.02 ± 0.92 ^a^	31.48 ± 2.01 ^c^	53.71 ± 0.57 ^b^	28.70 ± 1.30 ^cd^
Water Weight (g)	292.56 ± 19.77 ^a^	143.23 ± 15.29 ^c^	128.39 ± 9.40 ^c^	84.84 ± 10.20 ^c^	223.1 ± 10.99 ^b^	104.13 ± 5.98 ^c^	140.97 ± 15.37 ^c^	90.38 ± 13.41 ^c^
Root DW (g)	3.98 ± 0.43	1.83 ± 0.19	2.50 ± 0.49	1.08 ± 0.13	3.23 ± 0.34	1.56 ± 0.10	2.8 ± 0.27	1.34 ± 0.14

The table presents the mean thickness (µm, n = 12) of the cuticle and total leaf, along with the cuticle-to-leaf thickness ratio, for four genotypes under shaded and non-shaded conditions. Additional parameters include shoot fresh weight (Shoot FW), shoot dry weight (Shoot DW), root dry weight (Root DW), and water weight. Means within rows followed by different letters are significantly different according to Tukey’s HSD test (*p* ≤ 0.05).

## Data Availability

All data is presented in the manuscript and Appendix A.

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
