# Peer review of "Investigating Morphological and Physiological Responses to Stress in Begonia semperflorens"

_ijms, 2025, doi:10.3390/ijms26083514_

Round 1
Reviewer 1 Report
Comments and Suggestions for Authors
This research revealed significant genotype-dependent differences of morphological and physiological responses in stress responses in Begonia semperflorens. The results are interesting and provide critical insights for breeding programs focused on improving the resilience of wax begonias, supporting the development of heat- and light-tolerant cultivars for sustainable production in stress-prone environments. The manuscript can be considered after minor revisions. The detailed comments are as follows.
- In Part B of Figure 4, Anthocyanin Content, the height of the column “OPGC5104” and “Sprint White” are too low to see clearly. The authors can adjust this figure to make it clearer.
- Why did you select these Morphological and Physiological traits? Are these of special importance to stress responses?
Author Response
We greatly appreciate the reviewer’s positive feedback and insightful suggestions to enhance the clarity and justification of our manuscript. Below are our detailed responses to the specific comments provided:
Reviewer 1: "In Part B of Figure 4, Anthocyanin Content, the height of the columns 'OPGC5104' and 'Sprint White' are too low to see clearly. The authors can adjust this figure to make it clearer."
Response: We acknowledge that the anthocyanin levels in the genotypes 'OPGC5104' and 'Sprint White' are negligible (less than 0.1 absorbance units). We intentionally retained the existing graph scale to accurately represent the significant differences between genotypes, particularly highlighting the substantial increase observed in FB08-059 under non-shaded conditions. To clarify this point, we have added information on the negligible anthocyanin levels for OPGC5104 and 'Sprint White' in the text and Figure 4 caption. The edited text have been highlighted in the manuscript.
Reviewer 1: "Why did you select these Morphological and Physiological traits? Are these of special importance to stress responses?"
Response: We selected these morphological (cuticle thickness, leaf folding, stomatal density) and physiological (anthocyanin accumulation, chlorophyll content, gas exchange parameters, ROS accumulation, electrolyte leakage, and photosynthetic efficiency) traits based on their demonstrated importance in previous studies related to abiotic stress responses. Specifically:
- Cuticle thickness was evaluated because it serves as a critical physical barrier reducing water loss and protecting against excessive radiation.
- Leaf folding is a well-documented adaptive mechanism that reduces photodamage by minimizing direct light interception.
- Stomatal density directly influences gas exchange efficiency and water use, important factors under heat and high light stress.
- Anthocyanin accumulation is essential due to its dual roles in antioxidative defense and photoprotection.
- Chlorophyll content, gas exchange parameters (carbon assimilation, transpiration, stomatal conductance), and photosynthetic efficiency (Fv/Fm) are key indicators of photosynthetic performance and physiological stress.
- ROS accumulation and electrolyte leakage reflect oxidative stress and membrane integrity, crucial for assessing cellular health under stress conditions.
These traits collectively provide a comprehensive assessment of morphological and physiological adaptations relevant for stress resilience, informing targeted breeding strategies for improving heat and light tolerance in wax begonias.
Reviewer 2 Report
Comments and Suggestions for Authors
The paper entitled "nvestigating Morphological and Physiological Responses to Stress in Begonia semperflorens" study the responses to light and temprature but it study one stress the light (shaded or not shaded)
the introduction needs to be more clear for the effect of light in the physiological parameters (chlorophyll, pigments, antioxidant enzymes, protein and cabohydrates and soluble sugars)
materials : correlation statistical analysis needed to shoe the correlation among different genotypes under the stress (Principle component analysis and pearson correlation among different parameters)
Author Response
Reviewer 2: "The paper entitled 'Investigating Morphological and Physiological Responses to Stress in Begonia semperflorens' studies the responses to light and temperature, but it primarily focuses on one stress factor: light (shaded or non-shaded)."
Response: We appreciate the reviewer’s feedback on the scope of our study. While our experimental design categorizes plants under shaded and non-shaded conditions, the non-shaded treatment inherently introduces both high light intensity and elevated temperature stress, as confirmed by recorded environmental parameters. To clarify this point, we have included the temperature and humidity figures in the supplementary figure 2. We have also included in the discussion both light intensity and heat.
Reviewer 2: "The introduction needs to be clearer regarding the effect of light on physiological parameters (chlorophyll, pigments, antioxidant enzymes, proteins, carbohydrates, and soluble sugars)."
Response: We appreciate this suggestion and have revised the introduction to provide a more comprehensive background on how light intensity and heat influences key physiological parameters such as chlorophyll content, pigment accumulation, antioxidant enzyme activity, and metabolic adjustments (e.g., protein, carbohydrate, and soluble sugar levels). This will include relevant citations to support the physiological basis of light stress responses in plants. The edited section have been highlighted in the manuscript.
Reviewer 2: " correlation statistical analysis needed to shoe the correlation among different genotypes under the stress (Principle component analysis and pearson correlation among different parameters)”
We appreciate the reviewer's suggestion to include correlation statistical analysis, such as Principal Component Analysis (PCA) and Pearson correlation, to assess relationships among different parameters and genotypes under stress conditions. However, after careful consideration, we have decided not to implement these additional statistical analyses for the following reasons:
- Study Design and Research Focus: Our study was designed to compare the morphological and physiological responses of different wax begonia genotypes to high light and heat stress. The primary focus is on genotype-specific adaptations rather than multivariate relationships among parameters. The statistical analyses employed (ANOVA and Tukey’s HSD test) effectively capture significant differences between genotypes and treatments, aligning with our study objectives.
- Data Structure and Sample Size: While PCA and Pearson correlation are powerful tools for exploring underlying relationships among multiple variables, their effectiveness depends on sample size and data structure. Given our experimental setup, including four genotypes under two stress conditions with multiple replicates, our dataset is optimized for direct comparative analysis rather than dimensionality reduction or correlation modeling.
- Interpretability and Relevance: The addition of PCA and Pearson correlation may introduce complexity without providing substantial new insights into genotype performance under stress. Our measured traits, including chlorophyll content, anthocyanin accumulation, ROS levels, and photosynthetic efficiency, are already analyzed in a way that clearly differentiates stress-tolerant and stress-sensitive genotypes. The statistical approach used sufficiently demonstrates these differences without requiring additional correlation-based interpretations.
- Consistency with Previous Studies: Many studies investigating stress responses in ornamental plants have successfully used ANOVA-based analyses to identify genotype-specific variations. Our approach aligns with these established methodologies and ensures that our results are directly comparable to existing literature.
Based on these considerations, we believe that our current statistical analyses adequately address the research objectives and provide a clear, rigorous comparison of genotype responses under stress conditions. However, we acknowledge the value of PCA and correlation analysis in broader experimental designs and will consider incorporating them in future studies that focus specifically on trait interrelationships and multivariate data exploration.
Reviewer 3 Report
Comments and Suggestions for Authors
Comments to the manuscript entitled ‘nvestigating Morphological and Physiological Responses to Stress in Begonia semperflorens’ (ijms-3506251) written by Heqiang Huo, Julian Ginori, Chi Dinh Nguyen, Sandra Wilson, Zhanao Deng.
In reviewed manuscript authors studied response of four Begonia semperflorens genotypes to light and temperature stress. Subject of the research appears interesting and there are no doubts that experiments were executed carefully. However, in my opinion, the manuscript does not comply with the aim and scope of International Journal of Molecular Sciences. The authors only describe the results of morphological and very simple physiological measurements (with the exception of gas exchange measurements). They do not present any research results related to the molecular level (and the journal's Aims & Scope clearly states: ‘The International Journal of Molecular Sciences provides an advanced forum for molecular studies in biology and chemistry, with a strong emphasis on molecular biology and molecular medicine"). Furthermore, the results obtained only confirm the current state of knowledge on plant responses to the stressors studied.
Author Response
Reviewer 3: “In reviewed manuscript authors studied response of four Begonia semperflorens genotypes to light and temperature stress. Subject of the research appears interesting and there are no doubts that experiments were executed carefully. However, in my opinion, the manuscript does not comply with the aim and scope of International Journal of Molecular Sciences. The authors only describe the results of morphological and very simple physiological measurements (with the exception of gas exchange measurements). They do not present any research results related to the molecular level (and the journal's Aims & Scope clearly states: ‘The International Journal of Molecular Sciences provides an advanced forum for molecular studies in biology and chemistry, with a strong emphasis on molecular biology and molecular medicine"). Furthermore, the results obtained only confirm the current state of knowledge on plant responses to the stressors studied.”
We sincerely appreciate the reviewer’s thoughtful feedback and recognition of our study’s careful execution. Our manuscript, submitted to the Special Issue Plant Responses to Abiotic and Biotic Stresses in the International Journal of Molecular Sciences, aligns with the issue’s focus on physiological mechanisms of plant responses to abiotic stress. While our study primarily examines morphological and physiological responses, these traits serve as well-established indicators of underlying molecular mechanisms. Physiological markers such as chlorophyll degradation, anthocyanin accumulation, and oxidative stress responses reflect the biochemical and genetic processes regulating plant adaptation to stress. As many molecular studies rely on physiological and morphological data to validate gene function and metabolic pathways, our findings provide a critical bridge between plant physiology and molecular biology.
Furthermore, our study offers genotype-specific insights into stress tolerance in Begonia semperflorens, a species with limited available data on stress physiology. By identifying physiological traits associated with stress resilience, our research informs both breeding strategies and horticultural management practices. This knowledge is essential for developing more resilient cultivars capable of withstanding changing environmental conditions. Although we do not directly analyze molecular mechanisms, our findings establish a foundation for future transcriptomic and proteomic investigations that can further elucidate the genetic and biochemical pathways underlying stress responses. Given that the International Journal of Molecular Sciences includes plant physiology within its thematic scope, our study provides quantifiable physiological data essential for understanding stress adaptation, making it highly relevant to the journal’s focus.